# Inflammatory Bowel Disease: Role of Vagus Nerve Stimulation

**DOI:** 10.3390/jcm11195690

**Published:** 2022-09-26

**Authors:** Riccardo Fornaro, Giovanni Clemente Actis, Gian Paolo Caviglia, Demis Pitoni, Davide Giuseppe Ribaldone

**Affiliations:** 1Department of Neurosurgery, University Hospital “Maggiore Della Carità”, 28100 Novara, Italy; 2The Medical Center Practice Office, 10100 Turin, Italy; 3Department of Medical Sciences, Division of Gastroenterology, University of Torino, 10126 Torino, Italy

**Keywords:** ulcerative colitis, Crohn’s disease, acetyl-choline, IL-1β, corticotropin-releasing factor, central autonomic network, cholinergic anti-inflammatory pathway, α7 nicotinic ACh receptor, Cyberonics

## Abstract

Vagus nerve stimulation (VNS) is an accepted therapy for the treatment of refractory forms of epilepsy and depression. The brain–gut axis is increasingly being studied as a possible etiological factor of chronic inflammatory diseases, including inflammatory bowel diseases (IBD). A significant percentage of IBD patients lose response to treatments or experience side effects. In this perspective, VNS has shown the first efficacy data. The aim of this narrative review is to underline the biological plausibility of the use of VNS in patients affected by IBD, collect all clinical data in the literature, and hypothesize a target IBD population on which to focus the next clinical study.

## 1. Introduction

Inflammatory bowel disease (IBD) is an organic disorder that is typically categorized into CD and UC. While UC exclusively affects the rectum and the colon, CD can affect the entire digestive tract, from the mouth to the anus. IBD begins early in life (between 15 and 30 years) and progresses through flare-ups alternated with remissions of varying lengths. Genetic, immunologic, viral, and environmental variables all play a role in the complex pathophysiology of IBD [1]. Despite the presence of more and more new mechanisms of action, we are far from being able to cure these diseases. Furthermore, such drugs must be taken for a long time (if not for life), leading many patients to search for non-drug alternatives. In this review, we want to explore the potential of the vagus nerve (VN) and its stimulation as a possible new therapeutic weapon in IBD.

The vagus nerve (VN), an X pair of cranial nerves (also called the pneumogastric nerve), arises from the brain stem (including the pons, medulla oblongata, and midbrain), leaves the medulla oblongata, and travels down the chest and abdomen, passing through the jugular foramen.

The VN provides parasympathetic nerves to the digestive tract and pancreas, including 70–80% of sensory fibers; it is defined as a mixed sensory–motor nerve [2]. The system mainly recognizes two ganglia: the superior (jugular), which provides general sensation, and the inferior, which sorts the visceral and special sensation. In simplified anatomical–functional terms, the system recognizes efferent and afferent fibers. The former originates from the medullary dorsal motor nucleus and innervates the digestive tract from the esophagus to the splenic flexure, opposing the left colon and rectum, innervated by the S2-S4 sacral parasympathetic nucleus. This second-order neuron, a true “second brain” or “gut brain”, is a crucial component of the enteric (or intrinsic) neural system in the digestive tract, ensuring the motor and secretory autonomy of the digestive tract [3], with direct interaction with mast cells in the gut mucosa [4]. Notably, these efferent fibers are not anatomically connected with the intestinal lamina propria but are so with enteric neurons; these, in turn, can release acetyl-choline (ACh), a swinging modulator of muscarinic and/or nicotinic receptors. The afferent fibers originate from the mucosa and the muscular masses of the digestive tract: the relevant cells belong to the nodose ganglia, conveying information to the nucleus of the solitary tract and the postrema area in close relationship with the dorsal motor nucleus, giving rise to the complex dorsal vagal. Autonomic, endocrine, and limbic responses all depend on this complex relationship. Relevant information is extended to the amygdala, hypothalamus, and cortex through the main hypothalamic–pituitary–adrenal arm (discussed below) [5].

## 2. The Afferent Vagus and the Hypothalamic–Pituitary–Adrenal Anti-Inflammatory Pathway

For the purpose of maintaining homeostasis, research from the 20th and 21st centuries has identified the neuroendocrine route as the activation signal for endocrine, neural, and behavioral responses, allowing varying levels of innate defense (e.g., non-acquired immunity as opposed to acquired immunity). The VN, a key node in the defensive path, is now considered a cytokine receptor, including interleukin (IL)-1, IL-6, and tumor necrosis factor (TNF)-α. The protagonists, in the last decades, of the most typical pictures of experimental and spontaneous septic shock, IL-1, IL-6, and TNF-α are the cytokines that are the protagonists of translational research.

In extreme analysis, the afferent vagal tracts exert a specific anti-inflammatory action based on the following points:The vagal afferents are equipped with IL-1β receptors at the paraganglia level [6].From here, the information is conveyed to the core of the solitary tract.Through the paraventricular nucleus of the hypothalamus, the information is then extended to populations of specific neurons that release CRF (corticotropin-releasing factor) [7].The final task of these neurons is, therefore, to favor the release of the pituitaryadrenocorticotropic hormone (ACTH), which, as known, can, in turn, mediate the production and release of adrenocorticoids from the adrenal gland, having established anti-inflammatory effects (hypothalamic–pituitary–adrenal axis, HPA) (Figure 1).

The in-vivo importance of this anti-inflammatory vagal pathway is demonstrated by the distorted response to inflammogenic stimuli exhibited by vagotomized animals. Through its afferents, the VN may identify microbial metabolites and transmit this information to the central autonomic network (CAN), which can subsequently respond appropriately or inappropriately to the intestinal input and the microbiota [8].

### The Vagus Nerve at the Microbiota–Gut–Brain Axis Interconnection

The human gut is home to 10*^13^* to 10*^14^* bacteria; it has 100 times more genes than our genome and much more than our body’s cells. The microbiota weighs roughly 1 kg in humans, which is equivalent to the weight of the human brain. The microbiota–gut–brain axis has been proven to be a communication pathway between the gut, the microbiota, and the brain [9,10]. Different gastrointestinal pathogenic disorders, such as IBD, are characterized by dysbiosis [11]. A crucial part of this microbiota–gut–brain axis is the VN. The microbiota’s interactions with digestive endocrine cells, which release serotonin and act on vagal afferents’ 5-HT3 receptors, can indirectly activate it [8]. By detecting small-chain fatty acids (SCFAs) and/or gut hormones, vagal chemoreceptors are likely engaged in the communication between the microbiota and the brain [12]. In fact, *Lactobacillus johnsonii* intraduodenal injections increased stomach VN activity [13]. Chronic *Lactobacillus rhamnosus* treatment in healthy mice caused changes in GABA brain expression that were greater in the cingulate cortex and smaller in the hippocampus, amygdala, and locus coeruleus. These mice also showed a decrease in stress-related corticosterone levels as well as anxiety- and depression-related behavior. After vagotomy, these effects were not seen [14].

## 3. The Anti-Inflammatory Cholinergic Vagal Pathway (CAIP: Cholinergic Anti-Inflammatory Pathway)

If the previously mentioned hypothalamic–pituitary pathway inhibits inflammation by an afferent mechanism, it might be balanced by an efferent pathway, described by Tracey’s group in 2003 and reappraised in 2015. In this model, we speak of an “inflammatory reflex”, a sort of diastaltic arch where afferent vagal fibers activate their corresponding efferent ways. The following concepts were derived from an animal model of septic shock after intravenous injections of lipopolysaccharide: the shock was inhibited by stimulation of a dissected vagal terminal. Subsequent studies then identified ACh as the mediator released by the dissected terminal, demonstrating, in this context, the property of ACh to inhibit the release of shock effectors (IL-1, TNF-α, etc.) through the binding of ACh itself with the α7 nicotinic ACh receptor (α-7-nAchR) on macrophages [15]. Through the activation of α-7-nAchR, parasympathetic innervation of the gut contributes to the neuroimmune control of the intestinal barrier. It interacts with innate immune cells, myenteric neurons, and resident macrophages that express α7nAChR while acting on enteroglial cells [16].

Incidentally, these data can have an important translational value for gastroenterologists. In fact, the known protective effect of smoking on ulcerative colitis (UC) and, by contrast, the serious relapses in ex-smokers can be explained by the anti-inflammatory effect resulting from the link between vagal ACh and intestinal macrophages [17]; on the other hand, the lack of this effect, indeed, the negative effect of smoking in Crohn’s disease (CD), could suggest a different immunological status of the competent cells in CD. By contrast, there is a link between vagotomy and subsequent CD, demonstrating the importance of VN integrity in CD prevention [18].

On the other hand, academic rather than clinical issues are those raised by questioning the way by which vagal terminals and intestinal macrophages come in contact. If the evidence is that vagal fibers do not “physically” contact intestinal immunocytes, the nature of these other cells is far from clear as well. Two theories, none of which currently prevail, argue that vagal fibers interact with special intestinal neuronic targets or that splenic sympathetic activation comes into play. Norepinephrine (NE), which binds to the two adrenergic receptors of splenic lymphocytes and releases ACh, is released from the distal end of the splenic nerve. ACh then prevents spleen macrophages from releasing TNF-α through α-7-nAchR [19]. The idea of a non-neural connection between the vagus and splenic sympathetic nerves was put forth by Martelli et al. [20]. They hypothesized that the essential α7-containing nicotinic receptor is placed not on the cell bodies but on the peripheral terminals of the splenic sympathetic nerves. These nerves produce NE in response to ACh from entering T-cells, which then inhibits the generation of TNF-α by splenic macrophages by acting on their β2 adrenergic receptors (Figure 2).

Intestinal macrophages are a diverse group of innate immune cells that not only play a significant part in host defense but also sustain the tissue in which they are found [21]. Depending on where they are located in the gut, tissue macrophages have varied roles, unique cell dynamics, and unique physical characteristics [22]. For instance, the lamina propria is home to the majority of the intestine’s macrophages. These cells, which are identified by the expression of the CX3CR1 receptor, are situated close to the intestinal epithelium layer, where they monitor the environment, phagocytose potentially harmful antigens [23], and support epithelial cell renewal by secreting a number of mediators [24]. The lamina propria macrophages can also develop tolerance to food antigens and prevent excessive inflammation against safe commensal microorganisms, thanks to the production of anti-inflammatory cytokine receptors such as IL-10 [25,26]. In contrast to the lamina propria, the muscularis externa contains macrophages that are arranged in a complex network near the myenteric plexus. The latter represents the intrinsic innervation of the intestine, also known as the enteric neural system, along with the submucosal plexus. Intense reciprocal cross-talk is created by the near proximity of immune cells, particularly muscularis externa macrophages, to nerve fibers in the gut wall. This cross-talk is regulated by a complex combination of neurotransmitters, cytokines, and hormones [27,28]. In a mouse model of colitis, the severity of colitis was made worse by the adoptive transfer of macrophages from vagotomized mice into macrophage colony-stimulating factor 1 (MCSF 1)-deficient animals, indicating the crucial function of macrophages in the vagal anti-inflammatory effect [29].

Regarding the anti-TNF action of the VN, this effect of the VN is indirect, through contact with neuronal nitric oxide synthase (nNOS), choline O-acetyltransferase (ChAT), and vasoactive intestinal peptide (VIP) enteric neurons situated within the bowel muscularis mucosae; these neurons have nerve ends proximal to the resident macrophages [30]. T-cells that have been activated suppress macrophages systemically: additionally, they might move and reduce inflammation in regions that are not innervated by the VN [31].

By reducing intestinal permeability and modifying local immunity, the VN, through the CAIP, could modify the intestinal microbiota [8].

## 4. Possible Role of Vagus Nerve Stimulation in Chronic Inflammatory Bowel Disease

The demonstration that the vagal system affects inflammation regulation has endorsed exploratory attempts at therapeutics. Neuromodulation suggests using devices to control the nervous system’s electrical activity in order to restore organ function and health, partially avoiding toxicity, collateral damage, and poor compliance [32]. Epileptic syndromes, depression, and chronic gastroenterological and rheumatological disorders were initially the main targeted fields. In the late 19th century, vagus nerve stimulation (VNS) was first employed to treat epilepsy. VNS is now authorized for the treatment of depression and refractory epilepsy. With rising efficacy of up to 10 years, VNS, used to treat drug-resistant epilepsy, drives a 50% reduction in seizure frequency and intensity in 40–60% of patients, demonstrating that this treatment is a slow-acting therapy [33].

In gastroenterology, clinicians have focused on inflammatory bowel disease (IBD). IBD is characterized by an imbalance of the autonomic nerve system, vagal dysfunction in UC [34], and sympathetic dysfunction in CD [35], which may contribute to its pathogenesis. High serum TNF-α levels and salivary cortisol levels were connected with low vagal tone in CD patients, supporting the idea that the HPA axis and the autonomic nervous system are out of balance [36].

Positive correlations between vagotomies and subsequent IBD have been found, and this is especially true for CD, which highlights the importance of VN integrity in the prevention of IBD [18]. Hence, it stands to reason that in those who are at risk, the degree of vagal tone is a good indicator of the emergence of an inflammatory illness. Vagal hypotonia, which, in turn, keeps this inflammatory state in place, can be caused by the systemic inflammation seen with IBD or other chronic inflammatory illnesses. Furthermore, due to its central effects, persistent inflammation can cause depression, which, in turn, might trigger an inflammatory flare-up of the illness [37,38].

In two early studies, Lindgren et al. measured the cardiac responses to tilt (acceleration and brake index) and deep breathing (E/I ratio) in 40 UC and 33 CD patients to assess autonomic nerve function. They discovered sympathetic dysfunction in CD and vagal dysfunction in UC [34,35]. An increasingly popular noninvasive test to evaluate autonomic function is heart rate variability (HRV). In 27 IBD patients who were in remission and 28 healthy controls, Mouzas et al. evaluated HRV. When compared to healthy controls, it was shown that UC and CD patients who were in remission appeared to have more vagal activity [39]. According to Ganguli et al. in 2007, patients with UC but not CD showed more sympathetic activity compared to controls [40]. In a recent study, CD patients in remission exhibited a considerably larger sympathetic–parasympathetic ratio, according to Zawadka-Kunikowska et al. [41].

Some vagal afferent fibers come into close touch with intestinal mucosal mast cells as they go to the tips of the jejunal villi. These findings offer the microanatomical foundation for mast cells in the gastrointestinal mucosa to communicate directly with the central nervous system [4]. It is interesting to note that the efferent VN interacts with enteric neurons instead of directly connecting to the gut’s resident macrophages. Therefore, enteric neurons rather than vagal efferent fibers directly mediate the vagal regulation of these intestinal macrophages [27,30]. Pro-inflammatory cytokines that stimulate vagal afferents cause the activation of vagal efferents, which prevents tissue macrophages from releasing these cytokines, including TNF and other pro-inflammatory cytokines such as IL-6 and IL-1β but not the anti-inflammatory cytokine IL-10 [42]. This is known as an inflammatory reflex. 

### 4.1. Implantation of the Vagal Stimulator

The VNS maneuver requires the implantation of a device by a surgeon experienced in the specific field [43]. The maneuver takes about 1 h. An electrode is wrapped around the left VN neck area at the carotid artery, tunneled under the skin, and connected to a pulse generator implanted under the skin of the left chest. For the stimulation, the left VN is usually chosen because it is not involved in the regulation of heart rhythm [44]. The device is initially started at 0.25 mAmp, then progressively increased according to the patient’s need and tolerance. Stimulation is intended as continuous but is alternated with ON–OFF phases. Direct VNS delivered by an implanted pulse generator is typically secure and well tolerated. Hoarseness, increased coughing, changes in voice/speech, discomfort, throat or larynx spasms, headache, sleeplessness, indigestion, and other adverse effects are possible with VNS used to treat epilepsy [43]. The most frequent of them, typically transient, are hoarseness, coughing, throat tickling, and shortness of breath.

The effectiveness of various therapies is significantly influenced by the frequency of stimulation for VN activation [45]. The most prevalent applications of high frequency (20–30 Hz) are in the treatment of epilepsy and depression. High frequency is traditionally thought to trigger vagal afferents. Lower frequency (1–10 Hz), on the other hand, is thought to activate vagal efferents, which have anti-inflammatory characteristics. The end point of neurostimulation therapy in IBD coincides with the activation of the cholinergic system/CAIP activator (see above) through low-frequency stimulation (1–10 Hz) of the efferent fibers. VNS likely activates both its afferent (which activates the HPA) and efferent (which activates the CAIP) fibers to exert its anti-inflammatory effects in IBD.

### 4.2. Clinical Data in Inflammatory Bowel Disease

An initial study of experimental 2,4,6-trinitrobenzene sulfonic acid (TNBS) colitis in rodents demonstrated that VNS was able to moderate inflammatory data and colic lesions [46].

A VNS study was recently performed in CD patients, for which the VNS was positioned in the therapeutic algorithm at the same level as an anti-TNF [47]. The majority of the patients (8/9) had active CD at the time of inclusion. Electrode Model 302 was implanted with bipolar pulse generator Model 102 (Cyberonics, Houston, TX, United States); 10 Hz, 500 s, 0.5 mA, 30 s ON, 5 min OFF, constantly were the stimulation parameters. Patients between the ages of 18 and 65 who had a Crohn’s disease activity index (CDAI) score between 220 and 450 (i.e., moderate or severe CD), with small bowel (ileum) and/or colonic CD, C-reactive protein (CRP) > 5 mg/L and/or fecal calprotectin >100 μg/g, as well as a Crohn’s disease endoscopic index of severity (CDEIS) score ≥ 7 (active) and had been diagnosed for more than 3 months, were treatment-naive, or had a stable treatment reference (two patients were failing azathioprine, while the remaining seven were treatment-naive) were included. At the time of inclusion, patients who were taking infliximab or another anti-TNF drug were ineligible. VNS was carried out constantly for a year. The first patient received an implant in April 2012 and the final one in March 2016. After three months of neurostimulation, due to a worsening of their condition, two individuals were taken out of the study, most likely as a result of the especially high inclusion scores for CDAI (>350), CDEIS (>14), and CRP (>88 mg/L), which implies that VNS is recommended for mild-to-moderate active CD due to its gradual effect, as seen for epilepsy [48]. The first patient got an ileo-cecal resection but decided to continue neurostimulation until the end of the research due to an initial positive result and drug treatment refusal; the second patient was excluded from the study after three months due to a deterioration of disease. Infliximab and azathioprine were used to treat the second patient, who likewise desired to continue using an active VNS. Five of the nine patients experienced a deep (clinical–biological–endoscopic) remission, with also restored vagal tone, after receiving VNS. Median CDAI passed from 264 to 88, median CRP from 7 to 3 mg/L, median calprotectin from 847 to 61 µg/g, and median CDEIS from 8 to 0. They noticed that the VNS effect did not occur right away but rather took at least three months to manifest: in patients with significant flares, VNS should not be utilized alone, at least in the first months. Three more paradigmatic pro-inflammatory cytokines in CD, IL6, IL12, and IL23 were likewise decreased following a 12-month VNS. It is interesting to note that VNS decreases the perception of abdominal pain. Except for one patient who had an excessively high vagal tone, the majority of the CD patients studied here had low vagal tones upon inclusion that were restored to normal in all patients after a 12-month VNS. Although VNS might have a placebo effect, clinical remission following maintenance treatment ranged from 12% to 20% under placebo at 12 months in clinical trials using anti-TNF medicines in CD [49]. With the typical slight side effect of hoarseness as its main manifestation, VNS was well tolerated. According to the authors, these data indicate that VNS is feasible in IBD, but the study needs to be expanded to allow for interpretation.

In a second pilot trial, D’Haens et al. examined the effects of VNS over a period of 16 weeks in 16 patients who had colonic or small intestinal CD with biologic resistant disease [50]. Patients had CDAI 220–450, calprotectin >200 µg/g, an endoscopic score of activity (SES-CD: Simple Endoscopic Scale for Crohn’s Disease) with a minimal ulcer score of 2 or 3 in at least one segment, a history of inadequate response and/or intolerance or adverse events to one or more TNF-α inhibitors (e.g., infliximab, adalimumab, or certolizumab pegol), and an 8-week washout of biologics (group 1) or concomitant biologic therapy (Group 2). One minute of daily stimulation was started two weeks after a VNS device was implanted; between weeks 4 and 6, this was extended to five minutes; in the event that CDAI remission was not reached by week 8, stimulation was increased to four times daily. CDAI decreased from 294 to 201, calprotectin 2974 to 590, and SED-CD from 22.3 to 17.5. Seven of the sixteen patients obtained a CDAI-70 response and four of the sixteen patients CDAI remission. In terms of side effects, one patient developed a postoperative infection due to the device.

In nine ileo-colonic CD patients, Kibleur et al. evaluated the effects of VNS on inflammation and brain activity. They found that 12 months of chronic VNS improved CDAI, fecal calprotectin, anxiety state, and vagal tone, which were associated with a decline in the electroencephalogram’s α frequency band [51].

### 4.3. Possible Inclusion Criteria of Future Studies

CD with extensive small bowel disease or less than 200 cm of small bowel due to surgery and refractory to all target therapies;CD with contraindications to target therapies (fragile patients, history of cancer, history of severe infections);Patients with CD who do not want to undergo target therapy or with a history of poor adherence to therapy;Steroid-dependent UC.

## 5. Conclusions

The vagal system can express a modulating action of inflammation through the afferent pathways of the hypothalamic–pituitary–adrenal axis as well as through the efferent pathways of the cholinergic CAIP complex. Attempts to exploit these properties by means of devices capable of stimulating the VN under controlled conditions have obviously turned, as a starting point, to the treatment of chronic colitis (IBD) and rheumatological diseases: in these pathologies, where patients are particularly afraid of the toxicity of prolonged immune therapies, the introduction of VNS in a dominant position is attractive. The data are encouraging, but they must come out of the anecdotal stage to arrive at controlled designs.

## Figures and Tables

**Figure 1 jcm-11-05690-f001:**
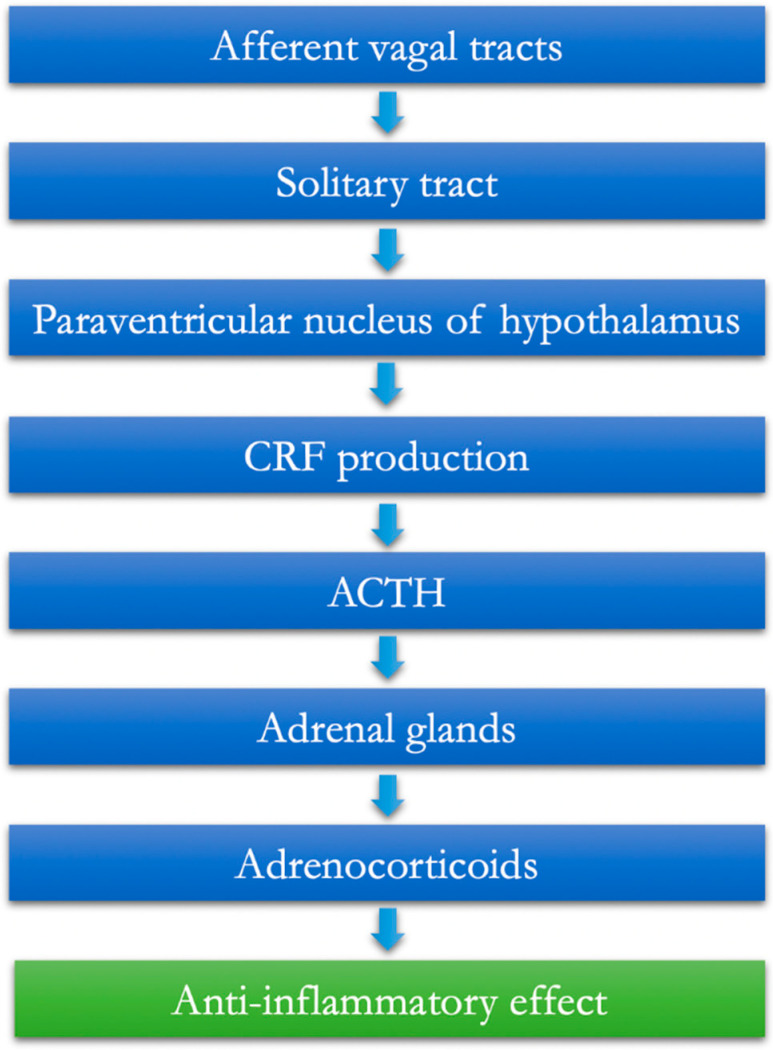
The afferent vagus and the hypothalamic–pituitary–adrenal anti-inflammatory pathway.

**Figure 2 jcm-11-05690-f002:**
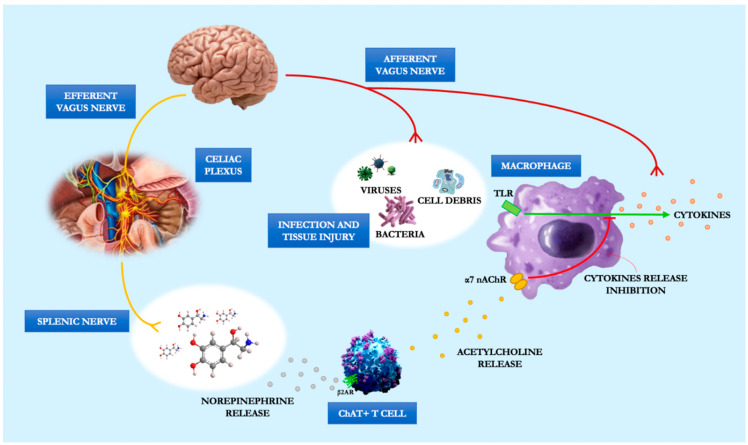
The anti-inflammatory cholinergic vagal pathway.

## Data Availability

Not applicable.

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
