# Peer review of "Inflammatory Bowel Disease: Role of Vagus Nerve Stimulation"

_jcm, 2022, doi:10.3390/jcm11195690_

Round 1

Reviewer 1 Report

In this paper, the author aimed to underline the biological plausibility of the use of VNS in patients affected by IBD, collect all clinical data in the literature, and hypothesize a target IBD population on which to focus the next clinical studies. The topic selection is very novel, which may help readers to have an in-depth understanding of this field. However, I have several comments on this paper.

1.     The relationship between vagus nerve and IBD is described in this paper, but the discussion is not deep enough. It is suggested to supplement some literature in this field in the manuscript, especially in part 4.

2.     The vagus nerve is a crucial component of the enteric (or intrinsic) neural system in the digestive tract, and a large number of immune cells exist in the lamina propria of the intestinal mucosa, which plays a key role in intestinal immunity and IBD. Descriptions of vagus nerve interactions with immune cells and their roles in IBD should be added. If the relevant literature is scarce, the data on the interaction of neurons or microglia with immune cells in other tissues, such as the nervous system, can be consulted.

3.     The interaction between vagus nerve and macrophages is listed in this paper, which is speculated to be important in IBD, but the references are all macrophages in spleen. Reference should be made to the roles of gut-resident macrophages. For example, it has been reported that a specific class of macrophages near the intestinal brachial plexus has been reported in the literature, we should focus on these cells subtypes instead of spleen macrophages.

4.     The references in some parts of the manuscript are missing, please add. For example, in Line 166, An initial study of experimental 2,4,6-trinitrobenzene sulfonic acid (TNBS) colitis rodents demonstrated that VNS was able to moderate inflammatory data and colic lesions. The reference should be added.

5.     Ther are some grammatical errors, for example, in Line 170, Eight of the 9 CD patients that were implanted by the authors had CD that was active at the time of inclusion. The structure of the sentence is not clear.

6.     Please modify the language. In some sections, it is extremely difficult to understand the meaning of the authors.

Author Response

In this paper, the author aimed to underline the biological plausibility of the use of VNS in patients affected by IBD, collect all clinical data in the literature, and hypothesize a target IBD population on which to focus the next clinical studies. The topic selection is very novel, which may help readers to have an in-depth understanding of this field. However, I have several comments on this paper.

Q1.     The relationship between vagus nerve and IBD is described in this paper, but the discussion is not deep enough. It is suggested to supplement some literature in this field in the manuscript, especially in part 4.

A1. Dear Reviewer, thank you very much for appreciating our paper.

According to your suggestion, we have investigated the link between the vagus nerve and IBD, citing the corresponding contributions present in the literature.

Q2.     The vagus nerve is a crucial component of the enteric (or intrinsic) neural system in the digestive tract, and a large number of immune cells exist in the lamina propria of the intestinal mucosa, which plays a key role in intestinal immunity and IBD. Descriptions of vagus nerve interactions with immune cells and their roles in IBD should be added. If the relevant literature is scarce, the data on the interaction of neurons or microglia with immune cells in other tissues, such as the nervous system, can be consulted.

A2. Thank you for your suggestion. We added the data on the interaction of neurons with immune cells in the tissues.

Q3.     The interaction between vagus nerve and macrophages is listed in this paper, which is speculated to be important in IBD, but the references are all macrophages in spleen. Reference should be made to the roles of gut-resident macrophages. For example, it has been reported that a specific class of macrophages near the intestinal brachial plexus has been reported in the literature, we should focus on these cells subtypes instead of spleen macrophages.

A3. Thank you for your suggestion. A focus on intestinal macrophages has been added.

Q4.     The references in some parts of the manuscript are missing, please add. For example, in Line 166, An initial study of experimental 2,4,6-trinitrobenzene sulfonic acid (TNBS) colitis rodents demonstrated that VNS was able to moderate inflammatory data and colic lesions. The reference should be added.

A4. We added the reference.

Q5.     Ther are some grammatical errors, for example, in Line 170, Eight of the 9 CD patients that were implanted by the authors had CD that was active at the time of inclusion. The structure of the sentence is not clear.

A5. We have rewritten the sentence.

Q6.     Please modify the language. In some sections, it is extremely difficult to understand the meaning of the authors.

A6. The language has been changed to be more understandable.

Reviewer 2 Report

The authors reviewed the role of vagus nerve stimulation in the treatment of inflammatory bowel disease. In introduction, they first described the anatomy of vagus nerve. Then they reviewed two anti-inflammatory pathways, the hypothalamic-pituitary-adrenal pathway and cholinergic vagal pathway. After that, they reviewed several important clinical studies as well as effective parameters. However, the author may consider reorganizing this manuscript and improving the language. For example, in introduction, it would be nice to include an overview of the manuscript, such as what will be discussed next. They may also consider moving the section 4 to the front of this manuscript so that the readers may have a better understanding of what CD is and the advantage and disadvantages of other types of treatments.

Author Response

The authors reviewed the role of vagus nerve stimulation in the treatment of inflammatory bowel disease. In introduction, they first described the anatomy of vagus nerve. Then they reviewed two anti-inflammatory pathways, the hypothalamic-pituitary-adrenal pathway and cholinergic vagal pathway. After that, they reviewed several important clinical studies as well as effective parameters. However, the author may consider reorganizing this manuscript and improving the language.

Q1. For example, in introduction, it would be nice to include an overview of the manuscript, such as what will be discussed next.

A1. Dear Reviewer, thank you for appreciating our manuscript.

We added an overview at the beginning of the introduction.

Q2. They may also consider moving the section 4 to the front of this manuscript so that the readers may have a better understanding of what CD is and the advantage and disadvantages of other types of treatments.

A2. Thank you for your suggestion. We added a paragraph at the beginning of the introduction to introduce what IBDs are and what unmet needs are.

Reviewer 3 Report

The article is well written organized and interesting. The author discussed possible use of VNS as alternative therapy for IBD individual.

1.       As mentioned in the manuscript as well that gut barrier is involved and there have a text where the author mentioned that microbial metabolites may interact with the central autonomic network (CAN). Keeping in mind the active role of gut barrier and microbial diversity in the gut, the author may can add a paragraph about linking microbial diversity; gut microbiome associated metabolites to VNS and subsequently IBD. As overall the text size is not that extended and so some new text could be added with addition of appropriate references.

2.      It would be great if some ideas are presented in graphical form and so will be easy for the reader to get a quick idea. Its highly recommended to add some figures.

3.      Check extra space in the text (Line-54).

Author Response

The article is well written organized and interesting. The author discussed possible use of VNS as alternative therapy for IBD individual.

Q1.       As mentioned in the manuscript as well that gut barrier is involved and there have a text where the author mentioned that microbial metabolites may interact with the central autonomic network (CAN). Keeping in mind the active role of gut barrier and microbial diversity in the gut, the author may can add a paragraph about linking microbial diversity; gut microbiome associated metabolites to VNS and subsequently IBD. As overall the text size is not that extended and so some new text could be added with addition of appropriate references.

A1. Dear Reviewer, thank you for appreciating our paper.

Thank you for your valuable suggestion. We added a paragraph entitled “The vagus nerve at the microbiota-gut-brain axis interconnection”.

Q2.      It would be great if some ideas are presented in graphical form and so will be easy for the reader to get a quick idea. Its highly recommended to add some figures.

A2. Dear Reviewer, thank you for your valuable suggestion. We have added Figure 1 which schematises the afferent vagus and the hypothalamic-pituitary-adrenal anti-inflammatory pathway and Figure 2 which illustrates the anti-inflammatory cholinergic vagal pathway.

Q3.      Check extra space in the text (Line-54).

A3. Sorry for the typo. We have corrected it.

Round 2

Reviewer 2 Report

Thanks! The authors have fully resolved my concerns.